# Cell Intrinsic and Extrinsic Mechanisms of Caveolin-1-Enhanced Metastasis

**DOI:** 10.3390/biom9080314

**Published:** 2019-07-29

**Authors:** America Campos, Renato Burgos-Ravanal, María Fernanda González, Ricardo Huilcaman, Lorena Lobos González, Andrew Frederick Geoffery Quest

**Affiliations:** 1Laboratorio de Comunicaciones Celulares, Centro de estudios en Ejercicio, Metabolismo y Cáncer (CEMC), Programa de Biología Celular y Molecular, Facultad de Medicina, Universidad de Chile, Santiago 8380453, Chile; 2Centro Avanzado para Estudios en Enfermedades Crónicas (ACCDIS), Santiago 8380453, Chile; 3Centro de Medicina Regenerativa, Facultad de Medicina, Universidad del Desarrollo-Clínica Alemana, Santiago 7590943, Chile

**Keywords:** caveolae, cholesterol transport, tumor suppressor, metastasis promoter, exosomes

## Abstract

Caveolin-1 (CAV1) is a scaffolding protein with a controversial role in cancer. This review will initially discuss earlier studies focused on the role as a tumor suppressor before elaborating subsequently on those relating to function of the protein as a promoter of metastasis. Different mechanisms are summarized illustrating how CAV1 promotes such traits upon expression in cancer cells (intrinsic mechanisms). More recently, it has become apparent that CAV1 is also a secreted protein that can be included into exosomes where it plays a significant role in determining cargo composition. Thus, we will also discuss how CAV1 containing exosomes from metastatic cells promote malignant traits in more benign recipient cells (extrinsic mechanisms). This ability appears, at least in part, attributable to the transfer of specific cargos present due to CAV1 rather than the transfer of CAV1 itself. The evolution of how our perception of CAV1 function has changed since its discovery is summarized graphically in a time line figure.

## 1. Introduction

Cancer, the second leading cause of death worldwide, arises from the transformation of normal cells into tumor cells in a sequence of events that generally progresses from a pre-cancerous lesion to a malignant tumor. This multifactorial pathological process, involves the acquisition of cancer “traits”, through the accumulation of genetic and epigenetic changes. The overall result is the loss of adequate communication between cells and their environment in ways that favor uncontrolled proliferation and tumor growth [1].

Cancer disease is initially characterized by uncontrolled growth of a primary tumor, which then is followed by spread of cancerous cells to surrounding and secondary tissues in a process referred to as metastasis. In doing so, cancer disease can affect almost any part of the body. Currently, it is estimated that metastasis is the primary cause of cancer mortality, and as such responsible for about 90% of cancer deaths. Cancer metastasis involves different steps, beginning with local invasion, followed by intravasation, survival in the circulation, extravasation, and finally colonization of the secondary site and growth at that site [2]. This process is promoted by genetic and epigenetic events that enhance the oncogenic potential of cells, prevent cell death, rewire metabolic pathways and bypass the immune surveillance system [1]. In addition, more recently evidence has accumulated implicating the release of extracellular vesicles (EVs) and particularly exosomes from cancer and stroma cells in these events [3].

In this review, we will discuss mechanisms relevant to development of metastatic disease with a focus on the role of EVs in this context. Additionally, we will center the discussion particularly on the role of a scaffolding protein called Caveolin-1 (CAV1) and how the protein contributes to this deadly process, through cell intrinsic mechanisms, as well as extrinsic pathways involving extracellular vesicles (see Figure 1).

## 2. Caveolins: Role in Physiological and Pathological Processes

Caveolin-1 (CAV1) is a 178 amino acid membrane protein, and the major structural protein of caveolae, 50–100 nm invaginations of the cell plasma membrane. These surface structures are found in many types of cells, such as adipocytes, epithelial cells, endothelial cells, fibroblasts, as well as smooth muscle cells, where they play important roles in membrane trafficking, determining membrane lipid composition and signal transduction. CAV1 contains several domains: An N-terminal domain (1–81 aa), a scaffolding domain (CSD, 82–101 aa), a hairpin-like transmembrane domain (102–134 aa), and a C-terminal domain (135–178 aa) [4]. The transmembrane domain contains two α-helices separated by a three-residue linker region including a proline (P110) that creates a ∼50° angle between the two α-helices. This allows CAV1 to adopt the hairpin-like topology mentioned above, such that both N- and C-termini face the interior of the cell [5] (see Figure 2). Thanks to efforts by different groups using circular dichroism spectroscopy and NMR techniques, the overall secondary structure of the CAV1 scaffolding and internal membrane domain have been elucidated; however, the tertiary structure of the full-length protein still remains elusive [4]. 

Two variants of CAV1 have been described, CAV1α (1–174 aa) and CAV1β (32–174 aa) shown in Figure 2; the latter is suggested to be generated by either alternative splicing or alternative initiation [6]. CAV1 is modified post-translationally by phosphorylation in the N-terminal region (Y14 and S80) and palmitoylation on three cysteine residues at the C-terminal by a reversible attachment of the 16-carbon acyl chain via a thioester bond [7]. These modifications are implicated in regulating steps in caveolae assembly, caveolae structure and signaling [6]. The development of CAV1(-/-) null mice permitted demonstrating the requirement for CAV1 in the formation of caveolae in vivo because all tissues analyzed in CAV1 null mice lacked these organelles [8,9]. Despite being widely accepted that CAV1 is necessary for the assembly of caveolae [10], it is also well known that other proteins are required. 

### 2.1. Caveolin-1 in Vesicular Transport

CAV1 is initially synthesized in the endoplasmic reticulum, where it is present as detergent-resistant oligomers [11,12]. Subsequently, once these are transported to the Golgi complex via COPII-dependent mechanisms, these oligomers associate with lipid domains enriched in cholesterol, which gives rise to the formation of a membrane-embedded complex containing around 15 to 25 caveolin molecules. These oligomeric CAV1 scaffolds are then transported to the plasma membrane where they associate with cavins and assemble into the stabilized coat structure characteristic of Caveolae [12]. Thus, cavins play an important role in defining structure and morphology of caveolae. In particular, absence of cavin-1 gives rise to cells which do not possess caveolae in their plasma membrane despite the presence of CAV1 [13]. Stability of all caveolae components is crucial, since the absence of any one of these elements (cholesterol, cavin, or CAV1) leads to loss of caveolae structures. Alternatively, caveolae are associated with enhanced stability of protein content. For instance, Hayer and collaborators showed that CAV1 turnover is accelerated by inhibiting caveolae assembly. In this case, CAV1 is targeted to endosomes, tagged with ubiquitin and captured within intraluminal vesicles of the multi-vesicle bodies which can fuse with the lysosomes [12]. When caveolae are correctly assembled, neither CAV1 nor cavin proteins undergo rapid turnover, but instead they can be internalized by endocytosis [12,14,15] or even engage in fusion and fission cycles with the cell membrane [12]. Initially, the proposed role for caveolae at the membrane was limited to the process of pinocytosis; however, with the development of new tools to investigate their function, their role as vesicular transporters was expanded to include transcytosis and endocytosis [16].

### 2.2. CAV1 in Cholesterol Homeostasis and Signal Transduction

Additionally, CAV1 and caveolae are implicated in cholesterol homeostasis. Rothberg and coworkers [17] were the first in recognizing the importance of cellular cholesterol for the assembly of caveolar structures, by showing that treating cells with cholesterol binding agents resulted in the flattening of caveolae. Moreover, cholesterol regulates CAV1 expression at a transcriptional level through two steroid regulatory binding elements in the CAV1 promoter [18,19]. This connection between CAV1 and cholesterol is even more intricate given that CAV1 can regulate cholesterol levels by modulating cellular influx and efflux [20]. Moreover, more recent evidence posits CAV1 as a key regulator of cholesterol levels in different subcellular organelles. For instance, in fibroblasts from CAV1 knock-out mice cholesterol accumulation in mitochondria is associated with dysfunction of that organelle [21].

Beyond the aforementioned functions, CAV1 and caveolae play an important role in signal transduction. Caveolae have been shown to concentrate a wide variety of signaling molecules, such as glycosyl phosphatidylinositol-linked proteins, H-Ras, heterotrimeric G protein subunits, Src family tyrosine kinases, Protein Kinase C (PKC) isoforms, and endothelial nitric oxide synthase (eNOS). These findings led to the “caveolae signaling hypothesis”, which suggests that caveolae function to compartmentalize signaling molecules and in this way regulate signal transduction [22]. Besides, many signaling molecules were proposed to directly interact with CAV1 via the caveolin-scaffolding domain (CSD). Often these interactions were shown to inhibit the respective proteins, as is the case for eNOS [23], epidermal growth factor receptor (EGFR) [24] and Src tyrosine kinases [25], among others. CAV1 contains a phosphorylable tyrosine residue (Y14), a target site for the non-receptor tyrosine kinases, including Src, Fyn, and c-Abl in response to a large number of stimuli [26,27,28]. Particularly relevant in this context, is that phosphorylation at this site has been widely associated with events important in cell migration [29]. Another site of phosphorylation is serine-80 (S80), an amino acid implicated in cholesterol transport as part of the cholesterol recognition/interaction amino acid consensus (CRAC) domain that is also implicated in retention of the protein in the endoplasmic reticulum (ER), and regulation of the secretion of the protein [30,31,32]. Of note, CAV1 is a ubiquitous protein that is present in addition to the plasma membrane, in a large number of intracellular compartments, including the ER, Golgi, mitochondria, endosomes, lipid droplets, and even in the nuclear membrane [33]. 

Regarding cholesterol metabolism, CAV1 has been found in mitochondria-associated membranes (MAMs), the physical association between the ER and mitochondria. Newly synthesized CAV1 in the ER binds to cholesterol and then exits ER and is transported to the Golgi apparatus. This immediate removal from the ER reduces cholesterol levels in general and also in MAMs. Thus, in the absence of CAV1, cholesterol accumulates in MAMs and mitochondrial membranes causing mitochondrial dysfunction and apoptosis [21,34]. In relation to autophagy, CAV1 appears to have an inhibitory role in hepatocellular carcinoma (HCC) since autophagy markers such as ATG5, Beclin-1, and LC3II were upregulated in HCCLM3-shCAV1 cells compared to mock cells [35]. In colorectal cancer, CAV1 depletion induces autophagy, in a p53-dependent manner [36]. Furthermore, CAV1 regulates autophagy in thyroid follicular cells [37]. These observations raise the specter that CAV1 at sites other than the plasma membrane is relevant to cell function and disease, although this remains an understudied area of research. 

### 2.3. Dual Role of CAV1 in Cancer

Given the wide variety of cellular processes that are modulated by CAV1 (caveolae), it is not surprising that this protein is involved in the development of many pathologies, such as cardiac hypertrophy, heart failure, fibrosis diseases, insulin resistance [38] and age-related diseases, such as atherosclerosis, osteoarthritis, pulmonary emphysema [39], and particularly cancer and metastatic disease [29]. CAV1 is a protein that has been ascribed a dual role in cancer, depending on cancer type and stage. In early stages of disease, CAV1 is proposed to function predominantly as a tumor suppressor, whereas at later stages, CAV1 expression is linked more to tumor progression and metastasis [29,40,41]. The first finding implicating CAV1 in cell transformation was the discovery that CAV1 is highly tyrosine phosphorylated in Rous sarcoma virus-transformed fibroblasts [42]. Subsequent experiments showed that protein levels of CAV1 are down regulated in oncogene transformed fibroblasts, and the reduction in CAV1 correlated with the increased size of colonies formed by these transformed cells when grown in soft agar [43]. Moreover, re-expression of CAV1 was able to revert the transformed phenotype and prevent anchorage independent growth [44]. These observations gave rise to the “oncosuppressor” hypothesis, suggesting that CAV1 functions as a tumor suppressor. Importantly, the role for CAV1 in tumor suppression is not based exclusively on in vitro experiments. CAV1 expression is known to be reduced in a number of human cancers, such as lung [45], mammary [46], colon [47,48], as well as ovarian [49], sarcoma [50], including osteosarcoma [51], and glioblastoma [52]. In view of such evidence, it is clear that CAV1 displays characteristics of a tumor suppressor, particularly in early stages of tumor development, although the precise mechanisms are often not well-defined. 

However, in subsequent years it became clear that the role of CAV1 in cancer is far more complicated. Particularly in later stages of the disease, expression of the protein increases and can favor the development of cellular characteristics related to enhanced malignancy, including multi-drug resistance and metastasis [40,41,53]. In prostate, CAV1 is not present in normal tissue, but expression increases along with the development of cancer in mouse and human models [54] and in vitro, CAV1 promotes metastatic features of prostate cancer cells [55]. In patients, expression of CAV1 in prostate tumors correlates with elevated metastatic potential and poor survival [56]. Similar observations have also been made for CAV1 expression during the development of melanoma [57,58,59]. In thyroid cancer, elevated expression of CAV1 and EGFR combined with the BRAF V600E mutation are associated with more aggressive lesions and thus may be useful for diagnosis [60]. Taken together, this evidence led to a model suggesting that CAV1 plays a dual role in cancer, behaving as a tumor suppressor in early stages of cancer, but as a tumor promoter in advanced and metastatic stages [40,41,53]. 

How CAV1 can develop such distinct functions in cancer cells represents an important and challenging area of research. Results from our laboratory suggest that this switch is linked, at least in part, to the presence or absence of E-cadherin in cancer cells. Our findings in colon cancer cells expressing E-cadherin indicate that CAV1 promotes the sequestration of β-catenin to the plasma membrane and thereby prevents β-catenin/(T-cell factor) Tcf-Lef dependent transcription of survivin and cyclo-oxygenase-2 (COX-2), both implicated in tumor development, progression, inflammation and angiogenesis [61,62,63,64]. Moreover, in murine melanoma B16F10 cells expressing low levels of CAV1 and E-cadherin levels, transfection with plasmids encoding CAV1, E-cadherin, or both proteins revealed that expression of either protein individually decreased the ability of B16F10 cells to form subcutaneous tumors in syngeneic C57BL/6 mice. Combined expression of the two proteins, completely abolished tumor formation by B16F10 cells. Intriguingly, metastasis to the lung following intravenous injection of cells into the tail vein was increased by CAV1 expression alone, while co-expression with E-cadherin ablated the ability of CAV1 to enhance melanoma migration in vitro and metastasis in vivo. These results suggest that E-cadherin determines CAV1 tumor suppression or metastasis enhancing function in melanoma cells [57]. 

The metastatic potential of tumor cells is determined by their ability to migrate and invade in response to changes in the extracellular matrix surrounding the tumor [53,65]. CAV1 controls cell mobility by interacting with the cytoskeleton and regulating cell interactions with the extracellular matrix [66]. In the human metastatic breast cancer cell line, MDA-MB-231, knock-down of CAV1 using a “small hairpin” leads to reduced migration, polarization, and focal adhesion turnover compared to MDA-MB-231 control cells [67]. CAV1 also regulates the degradation of the matrix by regulating the activity of matrix metalloproteinases (MMPs), which are proteolytic enzymes required for cell invasion [65]. 

More recently, a strong connection has been established between CAV1, cell migration, and metastasis [29]. CAV1 phosphorylation on Y14 is required for the control of pathways associated with cell migration [67,68,69,70,71]. Although mechanisms by which CAV1 promotes the migratory phenotype appear to depend on the cell type under study. For instance, in fibroblasts CAV1 promotes cell adhesion via RhoA activation and also accumulates at the rear of cells where it enhances calcium-signaling events [72,73]. Alternatively, in metastatic colon, breast cancer, and melanoma cells (lacking E-cadherin), CAV1 does not polarize during migration and presence of the protein following Y14 phosphorylation is associated with activation of a novel CAV1-Rab5-Rac1 signaling axis [29,74]. Thus, while Y14 phosphorylation appears to represent a common denominator to how CAV1 promotes migration and invasion, the pathways by which this modification triggers such responses varies depending on the cellular context.

### 2.4. Role of CAV1 in Metastatic Disease

Metastasis, as stated, is responsible for as almost 90% of cancer-related deaths and can be divided into different stages. Malignant cells from the primary tumor must first infiltrate the surrounding parenchyma and enter into the circulation by intravasation, survival in the circulation (hematogenous and/or lymphatic) where these disseminated tumor cells travel to distant sites, then extravasate, and finally colonize and metastasize target tissues. Following a process of adaptation, the preliminary micrometastatic cell mass grows into macroscopic metastatic nodules. This process is also referred to as the colonization of a target organ. The progression of metastasis is variable among different cancer types, and also, sometimes includes a varying period of latency. Each one of these steps is characterized by specific phenotypic features of the tumor cell, as well as interactions with the surrounding microenvironment and the immune system [2]. 

For metastasis to occur, cancer cells must acquire the ability to migrate and invade. Acquisition of these traits allows cells to degrade the surrounding extracellular matrix (ECM) and move towards blood and/or lymphatic vessels. In this context, evidence suggests a crucial role for CAV1 in different metastasis-related processes, such as epithelial to mesenchymal transition (EMT), cellular migration, ECM degradation, and angiogenesis, as summarized in Table 1 [29]. 

However, the role of CAV1 is complex and can also vary in metastasis and thus, be considered controversial, since depending on the context different signaling pathways are activated, as summarized below.

### 2.5. Downstream Signaling of CAV1 in Advanced Cancer

EMT is critical in cancer progression and metastasis, and involves the downregulation of epithelial markers (e.g., E-cadherin and γ-catenin), upregulation of mesenchymal markers (e.g., vimentin, fibronectin and N-cadherin) and transcription factors (e.g., Snail and Slug), which in conjunction enhance invasion, migration, and the acquisition of stem cell-like properties of cancer cells [90]. Another important feature of EMT is the upregulation of MMPs, which aid in the process of invasion. In addition, there is evidence that CAV1 is implicated in several aspects of EMT, thus driving cancer progression. For example, CAV1 is up-regulated after induction of EMT and upon expression enhances cancer cell adhesion [91]. Furthermore, CAV1 knockdown inhibits invasion and migration of BT474 breast cancer cells, upregulates E-cadherin and downregulates MMP-2, MMP-9 and MMP-1. Together, these results suggest that the inhibition of migration and invasion of BT474 cells following knockdown of CAV1 expression is attributable to the upregulation of E-cadherin and downregulation of MMPs [89]. Expression of CAV1 in SUM149 cells (an inflammatory breast cancer model) increased the invasive potential of these cells via activation of the Akt1 pathway, which phosphorylates the RhoC GTPase, a key player in microtubule and microfilament regulation, to promote cell adhesion and migration [87]. Díaz and collaborators showed that CAV1 can recruit p85α (a Rab5 GAP) and thus preclude p85α-mediated Rab5 inactivation [92]. Activation of Rab5 in turn increases the activity of Rac1 to enhance migration and invasion of breast and colon cancer cells, as well as melanoma cells. 

Upon detachment from the surrounding ECM, normal cells suffer a type of programmed cell death called anoikis. However, metastatic tumor cells are able to avoid anoikis upon detachment, which allows them then to invade other organs. CAV1 also participates in breast cancer metastasis by suppressing this process. When MDA-MB-231 cells detach from the ECM and enter the blood stream, CAV1 levels increase and promote resistance to anoikis by inactivating caspase-8 [76]. Similarly, CAV1 confers anoikis resistance to MDA-MB-231 cells by activation of PI3K/AKT and MEK/ERK pathways, as well as Integrin β1– focal adhesion kinase (FAK) signaling [93]. In another study, CAV1 phosphorylation enhanced HMGB1 secretion to the extracellular matrix, which activates toll-like receptor-4 (TLR4) signaling, nuclear factor kappa-light chain-enhancer of activated B cells (NF-κB) phosphorylation along with the upregulation of Snail and Twist, as well as MMP2 activation [94]. 

In hepatocellular carcinoma, CAV1 is over-expressed as mentioned previously and promotes cell motility and invasion by inducing EMT. Overexpression of CAV1 decreases E-cadherin expression and increases expression of N-cadherin, Fibronectin, and Vimentin, which are changes that are typically associated with the EMT process. Additionally, overexpression of CAV1 increased cell migration and invasion in these cells [95]. For melanoma, increased CAV1 expression was associated with disease progression by favoring migration, invasion and metastasis [57,58,96]. In lung adenocarcinoma cells, CAV1 expression is enough to promote filopodia formation, cell migration and increase metastatic potential [97]. 

In order for cells to migrate, the formation of focal adhesions is required at these contact sites between the cells and the ECM. CAV1 phosphorylation on tyrosine participates in the localization and stabilization of FAK in focal adhesions, an essential kinase that recruits p130Cas and paxillin, thereby promoting focal contact stability and subsequently focal adhesion turnover, which is essential to allow cells to move forward in the desired direction [69]. Increased FAK stability, migration, and invasion attributed to CAV1 presence involves the Src/Rho/Rho-associated kinase (ROCK) signaling axis and CAV1 phosphorylation on Y14 functions as an effector of Rho/ROCK signaling promoting tumor progression and metastasis [68]. Moreover, regarding the tumor-promoting role of CAV1 in absence of E-cadherin, downstream targets of CAV1 may contribute in promoting this role, specifically Rac1, whose activity increases significantly in presence of CAV1 and has also been held responsible for the regulation of several cellular behaviors, including cell migration and invasion [29,98,99]. In this regard, CAV1 overexpression in B16-F10 cells promotes migration, polarization and focal adhesion turnover, in a sequence of events that involves phosphorylation of tyrosine-14 along with Rac-1 activation [67]. Alternatively, depletion of CAV1 leads to focal adhesion disorganization and reduced cell migration [100,101]. Although important in cell migration, CAV1 function depends on the cell type. Given that there are several intracellular CAV1 pools, these may also be important in defining the seemingly different roles of CAV1 [29,71]. 

CAV1 is also involved in endocytosis, a process that controls the availability and turnover of cell surface molecules, as well as the ability of a cell to respond to certain extracellular stimuli. Thus, CAV1 is implicated in the turnover of growth factor receptors, like the EGFR [102], ECM binding proteins, like the β1 integrin [103], as well as the turnover of E-cadherin, known to be highly relevant to metastasis [104]. In order for tumor cells to metastasize, they need to remodel the cell-ECM interactions, in a way that allows tumor cells to escape from the original site to the bloodstream and colonize a new site. This can be achieved by degradation of ECM via release of MMPs, plasmin, and other proteases, as well as by internalization of ECM proteins and their subsequent intracellular degradation via lysosomes. There is evidence that both processes are modulated by CAV1. For example, CAV1 has been shown to organize ECM remodeling by coordinating proteolysis at sites adjacent to the apical membrane [66]. 

### 2.6. CAV1 in Preclinical Studies

In view of this information, CAV1 has also been mentioned as a potentially useful marker for diagnosis and treatment. For instance, in triple negative breast cancer (TNBC), the most aggressive type of breast cancer, no detectable targets are available compared to other subtypes. However, CAV1 does appear in a list of genes associated with Wnt/β-catenin signaling, which is used to distinguish mesenchymal subtype TNBC from the luminal subtype. Lehman and collaborators also described that the mesenchymal subtype may be targeted efficiently with specific inhibitors (e.g PI3K/mTOR and abl/src inhibitors). Such evidence can be taken to suggest that the detection of CAV1, may serve to identify which type of treatment will be more effective at eliminating specific subtypes of TNBC [105]. Along a different line, a recent study with the herbal adjuvant drug termed *Oldenlandia diffusa* (OD), which is used in traditional Chinese medicine to treat advanced-stage breast cancer patients, may inhibit the development of metastasis by diminishing CAV1 expression [106]. In vitro migration and invasion experiments using the highly metastatic breast cancer cell lines MDA-MB-231 and MDA-MB-453 revealed that reduced CAV1 levels due to OD treatment coincided with reduced migration/invasion by these cells, and that an overexpression of CAV1 attenuated the beneficial effects of OD in these cell lines. Extrapolation to the clinical setting suggests that elevated CAV1, often observed in advanced-stage cancers, can be successfully targeted with existing treatments to reduce the metastatic potential of tumor cells.

In summary, evidence is available implicating CAV1 as a protein that precludes as well as favors the acquisition of cancer cell traits associated with enhanced or reduced metastatic potential. However, a majority of the data available implicate the protein as displaying a pro-metastatic role. This notion is further supported by a considerable body of evidence suggesting that increased CAV1 favors experimental metastasis of tumor cells of varying origin, including those from prostate [55,107], pancreas [108], bladder [109], and melanomas [57,71]. As such, CAV1 may also have some potential in the diagnosis and as a therapeutic target in cancer disease.

## 3. Caveolin-1 Outside of the Cell: CAV1 as a Secretable Protein 

The evidence discussed in previous sections focusses largely on how CAV1 modulates cell function as an intracellular protein, be the site of action the plasma membrane or another location within the cells. However, a considerable amount of evidence now points towards the possibility that “extracellular” CAV1 may be particularly relevant in cancer cell metastasis.

The first report suggesting that CAV1 entered the secretory pathway was obtained in exocrine cells [110]. Anderson and coworkers reported on the secretion of CAV1 from pancreatic acinar cells and a transfected exocrine cell line, by the treatment with the secretagogue secretin, cholecystokinin, and dexamethasone. In addition, this report revealed that the secreted CAV1 co-fractionated with apolipoproteins, suggesting that the secreted protein may be associated with lipids. Subsequently, pituitary cells were also reported to secrete CAV1. However, unlike pancreatic acinar cells, CAV1 secretion was not regulated by secretagogues [111]. 

In the same year, Lisanti and colleagues employed a site-directed mutational approach to elucidate the functional contribution of phosphorylation at two highly conserved serine residues of CAV1. Mutation of Ser80 to alanine (S80A) precludes phosphorylation and targeted the protein to caveolae membranes; however, the protein was not secreted by pancreatic adenocarcinoma cells even following dexamethasone stimulation. Alternatively, substitution of Serine 80 by glutamate (S80E), which is a mutation that mimics chronic phosphorylation, lead to loss of CAV1 from caveolae and the ER in fibroblasts. In addition, CAV1(S80E) secretion was enhanced compared to wildtype CAV1 following dexamethasone treatment. These findings were taken to suggest that phosphorylation on S80 may regulate CAV1 binding to ER membranes and incorporation into the regulated secretory pathway [30]. 

### 3.1. Secretable CAV1 Promotes the Acquisition of Malignant Traits in Recipient Cells

Thompson and colleagues were the first to show that CAV1 is secreted by prostate cancer cells in a manner regulated by steroid hormones. CAV1 was detected in serum from patients with advanced prostate cancer and to a significantly lesser extent in normal subjects. In addition, they provided evidence for the functionality of CAV1 secreted by cells. CAV1-containing conditioned media (CM) from high passage CAV1-secreting, human prostate cancer LNCaP (LNCaPCAV1) cells augmented viability and clonal growth of low passage, CAV1-negative, LNCaP cells in vitro, and addition of CAV1-specific antibodies to the CM blocked this effect. Moreover, intraperitoneal injections of mice with these CAV1-specific antibodies suppressed the orthotopic growth and spontaneous metastasis of highly metastatic, CAV1-secreting mouse prostate cancer cells in vivo [107]. 

Expression of CAV1 in LNCaP cells increased cell proliferation and tumor growth in vivo. LNCaPCAV1 cells injected into one flank of nude mice promoted tumor growth of LNCaP cells (initially lacking CAV1) injected into the contralateral flank of the animal. Interestingly, the LNCaP tumors were positive for CAV1; however, presence of the protein was not attributable to infiltration by LNCaPCAV1 cells. Rather CAV1 was secreted by the LNCaPCAV1 tumors. Moreover, conditioned media (CM) from LNCaPCAV1 cells contained CAV1 associated with 15 to 30 nm lipoprotein particles. These results suggest that LNCaPCAV1 cells secrete CAV1-containing particles that stimulate tumor growth [112].

Interestingly, CAV1 was detected in the conditioned media (CM) of human melanoma cells expressing this protein. Functionally, a significant increase in invasion and migration of cells was observed in the presence of CAV1-containing CM. Moreover, addition of a polyclonal CAV1-specific antibody to the CM from human melanoma cell lines reduced cell migration and invasion. Taken together these data indicate that secreted CAV1 promotes invasion of melanoma cell lines [96]. 

Additionally, Ewing’s sarcoma (EWS) cells were found to secrete CAV1. Furthermore, these cells can take up the secreted protein and CM containing CAV1 increases the proliferation of EWS cells [113]. Thus, evidence available from several groups indicates that CAV1 is secreted by cancer cells and can promote the acquisition of traits associated with enhanced malignancy of such cells. In addition, other reports show that CAV1 is enriched in the endolysosomal compartment of human melanoma cells, where it is suggested to contribute to the regulation of some functions of tumor cells mediated by vesicles, such as cellular cannibalism, which generally refers to the capacity of a cell to engulf another smaller cell [114].

### 3.2. CAV1 Released in Extracellular Vesicles

Prostate epithelial cells produce prostasomes, defined vesicular organelles enriched in raft components that are secreted in the prostate fluid. The prostasomes isolated from the CM of PC-3 cells were shown to be 30 to 130 nm vesicles that contained CAV1 and CD63, an integral membrane protein found in multi-vesicular bodies/lysosomes and secretory granules [115]. Sawada and colleagues showed that CAV1 is present in the “matrix vesicle” fraction from osteoblasts. In addition, this study was the first to show that caveolae membrane fractions and matrix vesicles of MC3T3-E1 cells are similarly enriched in cholesterol and sphingomyelin [116]. 

CAV1-containing exosomes and melanoma cells lacking CAV1 were used to test whether low pH conditions, a hallmark of tumor malignancy, increase the intercellular incorporation of tumor-associated molecules through exosomes. Indeed, acidic pH conditions favored exosome-mediated delivery of CAV1 to less aggressive melanoma cells lacking CAV1. These data suggest that low pH conditions favor cell-to-cell exchange by exosomes and the transfer of characteristics associated with the more aggressive phenotype to less aggressive sibling cells [117].

Using an ELISA assay to analyze plasma samples, a significantly augmented ratio of exosomes containing tumor markers, including CAV1, was detected in melanoma patients compared to healthy individuals. Alternatively, the plasma levels of CAV1-containing exosomes decreased significantly in patients undergoing chemotherapy compared to patients that had not been treated at the time of sampling [59].

Di Vizio and colleagues described methods for the analysis of large oncosomes in the absence of vesicles < 1 µm. The sorting of CAV1-positive vesicles from the plasma of transgenic mice with autochthonous prostate tumors (TRAMP) showed that for this tumor model, microvesicles (>2 to 3 µm) larger than the exosome-sized particles could be detected and quantified in tissues and in circulation. These vesicles were present in greater quantities in the circulation of TRAMP mice with lymph node and lung metastases. These results suggested that CAV1-containing large oncosomes reached the circulation of mice with prostate cancer. Importantly, the presence of smaller CAV1–positive vesicles (<1 µm), which were also detected in the plasma, did not correlate with disease progression in this model [118].

Interestingly, recent studies revealed the existence of extracellular vesicle (EV)-mediated communication between different cell types within the adipose tissue. This phenomenon was evidenced upon generating an adipocyte-specific knockout of CAV1. Although the CAV1 gene was effectively knocked out in adipocytes, CAV1 protein remained abundantly detectable. By generating additional mouse models, it was then shown that neighboring endothelial cells (ECs) deliver CAV1-containing EVs to adipocytes in vivo [119].

### 3.3. CAV1-Containing EVs Promote Malignancy of Recipient Cells

In vitro hypoxia experiments using glioma cells combined with the analysis of patient samples revealed an increase in hypoxia-regulated mRNAs and proteins (including CAV1) in exosomes. The exosomes derived from hypoxic glioblastoma multiforme (GBM) cells increased autocrine secretion of growth factors and cytokines compared with exosomes isolated under normoxic conditions, and were able to activate the PI3K/AKT signaling, as well as increase the migration of endothelial cells [120].

Characterization of the proteome content by mass spectrometry of exosomes derived from three HCC cell lines and an immortalized hepatocyte line showed that only the exosomes derived from metastatic HCC cells contained CAV1. Notably, exosomes from the motile HCC cell lines augmented the migration and invasion of the non-motile, immortalized hepatocyte cell line [121].

More recent work from our laboratory showed that the CAV1-containing EVs from MDA-MB-231 breast cancer cells contain different protein cargos. EV analysis by proteomics revealed that some proteins related to biological adhesion, like tenascin and cysteine-rich angiogenic inducer 61 (Cyr61), were detected only in CAV1-containing EVs. Moreover, only incubation with the CAV1-containing EVs increased the migration and invasion of non-metastatic breast cancer cells. These observations are the first reported evidence that support the notion that the presence of CAV1 plays an active role in determining the cargo protein composition/biogenesis of EVs and that such differences in cargo have functional consequence in the recipient cells [122].

## 4. Extracellular Vesicles in Cancer

EVs are vesicles composed of a heterogeneous phospholipid bilayer, which are actively secreted by different kinds of mammalian cells, and particularly dividing cells [123]. They were initially identified as structures involved in the removal of cellular debris, as is the case for receptors by reticulocytes [124]. More recently; however, their role is considered far more complex. Specifically, in cancer, EV functions vary considerably and may include their use as biomarkers of disease [125], modulation of the immune response [126], changes in proliferation [127], modulation of non-tumor cells in the tumor microenvironment [128,129], as well as conditioning the metastatic niche [130]. More recent studies showed that these vesicles aid in driving cancer progression, since they facilitate communication by exchanging components ranging from nucleic acids to lipids and proteins, between the cells of origin and nearby cells, as well as distant tissues [131,132].

### 4.1. Types of Extracellular Vesicles

Essentially two mechanisms are considered most relevant for the formation of these EVs. One involves blebbing of the plasma membrane followed by membrane fission to release the vesicles [133], which are often referred to as microvesicles. However, depending on the cellular origin, such EVs have other names, including ectosomes, oncosomes, and platelet dust, among others. These EVs vary greatly in size and can be anywhere from 50 to 1000 nm or even larger.

The second origin of EVs is more intricate and currently considered of greater interest. As part of the endocytic pathway, vesicular structures are generated, referred to as multivesicular bodies (MVBs), that contain intraluminal vesicles (ILVs), generated by budding of the endosomal membrane towards the luminal space [131]. As mentioned previously, MVBs may fuse with lysosomes, which lead to degradation of MVB content (including CAV1; see Section 2.1); alternatively, they can also traffic to the plasma membrane, then fuse there and release the ILVs to the cell exterior, to yield what are commonly known as exosomes [134]. The size range of exosomes is more limited, being between 50 to 150 nm [131].

### 4.2. Biogenesis of Exosomes

One of the best characterized mechanisms involved in vesicle secretion is the endosomal sorting complex required for transport (ESCRT), a system that selects proteins by ubiquitination and segregates transmembrane proteins to microdomains of the endosomal membrane (ESCRT-0 and ESCRT-I). Others (ESCRT-II) participate in the recruitment to those domains of complexes that cause the deformation and fission of the ILV (ESCRT-III) [135]. These complexes require the assistance of accessory proteins, such as ALIX/AIP, tumor susceptibility gene 101 (tsg101) and vacuolar sorting protein 4 (VSP4), which facilitate the interaction of cargos with subunits of the ESCRT machinery and direct them to multivesicular bodies. Modification of the ESCRT complexes has a variety of different consequences for the genesis of EVs, but the specific responses vary in a cell type-specific manner. For example, the depletion of ESCRT elements, such as ALIX, reduces the production of vesicles by MCF7, but in HeLa cells [136] there is no clear response with respect to the production of exosomes.

These results indicate that EV production is not dependent on a single type of machinery and that there are other ESCRT-independent mechanisms involved in the genesis of extracellular vesicles. For instance, there is lipid-dependent machinery, which generates microdomains required for the genesis of intraluminal vesicles by favoring luminal bending of the MVB. Important in this context is the neutral sphingomyelinase, which processes sphingomyelin to generate ceramide, a conical lipid, which favors the deformation of the endosomal membrane [137]. The relevance of this enzyme is evidenced by showing that inhibition affects both exosome number and cargo [138,139]. In addition to ceramide, lipidomics analysis revealed that the membrane domains that give rise to the exosomes are enriched in lyso-phospholipids and cholesterol, both lipids which form raft-like structures that favor luminal bending of the MVB membranes [140,141].

Another ESCRT-independent mechanism involves endosomal microdomains rich in CD63 [140] and CD81 [142], tetraspanins that interact with cargoes destined for exosomes and whose functioning is not altered by eliminating components of the ESCRT machinery.

### 4.3. Exosomal Cargos

The content of EVs frequently is a reflection of the cell of origin and this aspect is important when considering exosomes for diagnosis. The ESCRT machinery is responsible for defining protein cargoes enriched in the exosomes. But in addition to proteins, exosomes also transport nucleic acids, as DNA [143] and different classes of RNAs, such as messenger RNA and non-coding RNAs (including micro-RNA (miRNA) and long non-coding RNA) [144].

It is still a matter of debate as to whether the transfer of nucleic acids to the exosomes is a passive event in which certain abundant RNAs accumulate in these vesicles, or rather whether there are selective mechanisms that facilitate the enrichment of certain RNAs. In favor of the latter possibility, some RNA binding proteins (RBP) have increased affinity for the ceramide-rich domains of multivesicular bodies, such as those destined for formation of exosomes [145]. These proteins recognize short sequences of nucleic acids that would allow targeting of these molecules to vesicles [146]. Other molecular machineries involved in the transfer of such material to exosomes include ESCRTs, miRNA-mediated sorting complex (miRISC), and protein argonaute-2 (AGO-2) [147].

### 4.4. EV Secretion

The secretion of EVs can occur in response to specific stimuli (fetal bovine serum, infection) [148,149], which is indicative of a regulated process. However, secretion is also observed in the absence of a stimulus, as a constitutive event [150]. In this scenario, the machinery of the endosomal pathway that determines the fate of multivesicular bodies must be considered. In this process, Rab GTPases are responsible for the trafficking of endosomal vesicles. One of the most important Rabs in the context of endosome secretion is Rab27 [151]. This Rab modifies both the interaction of the multivesicular bodies with the cell membrane and the secretion of EVs. Other Rab GTPases, such as Rab11 and Rab35, generally implicated in early endosome trafficking, are now also considered as regulatory molecules of exosome secretion [152,153]. Finally, for the fusion of multivesicular bodies with the cell membrane, the soluble N-ethylmaleimide-sensitive factor activating protein receptor (SNARE) proteins are important, given that they form complexes that facilitate organelle membrane fusion [154,155].

### 4.5. EV Uptake

Considering the importance attributed to EVs in cellular communication, at least three possibilities are considered relevant to the interaction with the recipient cells. The first possibility involves reciprocate recognition of surface molecules present on the vesicles and the cells. This interaction may suffice to activate the recipient cell without the necessity of delivering the intravesicular cargo [156]. However, fusion between vesicles and the recipient cell plasma membrane may also occur resulting in release of the encapsulated content into cells [157]. Additionally, EVs can be internalized by endocytosis, via either clathrin-dependent or -independent molecular machineries [158,159,160]. In these events participation of lipid-rafts and CAV1 has also been reported. For instance, manipulation of lipid-rafts, by diminishing cholesterol content using methyl-β-cyclodextrin (MβCD) or simvastatin, can also inhibit EV uptake. The function of CAV1 in the recipient cells is more debated, since in some cases the presence of CAV1 favors internalization [161], while in others CAV1 delays the internalization of exosomes [162].

After endocytosis or phagocytosis, EVs ultimately reach the MVBs of the recipient cell. Frequently, the final destination of such internalized structures is the lysosome; however, in some cases endocytosed vesicles gain access to the cytosol from MVBs by vesicle attachment with the endosomal membrane, an essential step for cytosolic proteins and nucleic acids to accomplish their function [163]. Other reports show that after the vesicles have been internalized, they make brief contact with the endoplasmic reticulum [164]. This is potentially of great importance, considering that in doing so vesicular mRNAs and miRNAs could gain direct access to the translation machinery of the recipient cell. In summary, the role of CAV1 in EVs is an interesting and still to be explored field. 

### 4.6. CAV1 Involved in EV Biogenesis and Protein Sorting

Phosphorylation of CAV1 on tyrosine 14 has more recently also been implicated in the internalization and subsequent redistribution of this protein to the late endosome or MVBs. Redistribution to this compartment is observed in the absence of cavin-1 (PTRF), an essential coat protein of caveolae [165], in mouse-derived embryonic fibroblasts (MEFs) and COS7 cells, or when these same cells were exposed to orthovanadate [166] or EDTA [167]. In addition, tyrosine-14 phosphorylation of CAV1 alone may not suffice. Indeed, N-terminal lysine residues of CAV1 (Figure 2) were identified as critical ubiquitin conjugation sites for sorting of this protein to other endocytic compartments by recruiting a specific type of AAA+ ATPase, termed VCP (valosin-containing protein, also known as p97/Cdc48) and its cofactor UBXD1 [168].

Additionally, UBXD1 in the presence of another VCP cofactor, Ankrd13, is capable of regulating VCP interaction with ubiquitinated CAV1 and; therefore, can drive CAV1 sorting to intraluminal vesicles in the endolysosomal compartment [168,169,170]. 

Although the sorting of CAV1 is thought to occur in the late endosome compartment or MVBs and lead to degradation at the lysosomal level [12], there are reports which show how stimuli, such as changes in cellular cholesterol levels, or an unbalanced lysosomal pH may promote the presence of CAV1 in MVBs without altering their degradation rate [171]. Thus, the latter observation points towards to the existence of alternative regulatory pathways for this type of CAV1 pool.

In addition to CAV1 phosphorylation on tyrosine 14 and its ubiquitination at specific N-terminal lysine residues, other post-translational modifications, such as palmitoylation, might be essential to dynamically modulate the balance of CAV1 between degradation or its secretion in EVs [172]. Additionally, in this regard, the presence of other scaffolding proteins, such as flotillin-1, have been reported to modulate the sorting of CAV1 into exosomes in PC3 prostate cancer cells [140]. Furthermore, inhibition of the PIKfyve kinase in PC3 cells not only leads to an increase in the rate of the secretion of exosomes, but also alters exosome composition, as reflected in a modest increase in CAV1 and Alix presence [173]. 

Accordingly, the question that arises is whether CAV1 might be able to promote segregation of other proteins into EVs. This may be possible, since results from our laboratory provide evidence indicating that CAV1-containing EVs from breast cancer cells are enriched in cell adhesion-related proteins, such as cysteine-rich angiogenic inducer 61 (Cyr61), S100A9, and tenascin [122]. On the other hand, for EVs obtained from the same metastatic breast cancer cell lines, where CAV1 was silenced, proteomic profiling and gene ontology analysis revealed preferential accumulation of proteins that specifically bind to DNA/RNA, such as histones, and chaperones supplementary material [122]. Thus, at least in this metastatic breast cancer cell model, CAV1 presence in EV preparations is associated with the accumulation of specific metastasis-related components.

### 4.7. CAV1-Containing EVs Transport Proteins Which Promote Malignant Traits in Recipient Cells

Beyond the question of changes in protein and lipid content of EVs/exosomes due to CAV1 presence, the key issue is whether these changes in composition lead to functional changes in, for instance, recipient cells. In this context relevant, CAV1 and CD59 are present together with delta-catenin in urinary EVs of prostate cancer patients, whereby delta-catenin was shown to be released to the extracellular matrix thanks to the presence of CAV1 and CD59 [174,175]. 

Taraska and collaborators described that EVs from metastatic breast cancer cells with high endogenous CAV1 levels are capable of stimulating migration in non-metastatic breast cancer cells such as MCF7 [176], which coincidently express relatively low levels of CAV1. However, these results did not attribute this characteristic to the selective presence of this protein. Alternatively, results from our research group revealed that T47-D cells lacking CAV1 increased their migration and invasion after being exposed to EVs derived from metastatic breast cancer cells with high endogenous levels of CAV1. Interestingly, and somewhat counterintuitively, CAV1-containing EVs also promoted the migration and invasion of breast cancer cells that already expressed high levels of CAV1 [122]. This may be also taken to suggest that it is not the sole presence of CAV1 in EVs that contributes to the observed biological effects, but that instead specific components are co-segregated with CAV1, such as the aforementioned cell adhesion-related proteins, which might be potentiating the observed biological effects in vitro. 

In this regard, Cyr61, a member of the CCN (connective, cysteine, nephroblastoma) family of matricellular proteins that also includes connective tissue growth factor (CTGF) [177,178] is of interest given that Cyr61 promotes the migration, growth, and invasion of tumor cells in gastric, breast, and ovarian cancers [179,180]. Specifically, reports described that the high Cyr61 expression promotes the dissemination of gastric cancer cells to the peritoneal cavity through α2β1 integrin [181]. Additionally, the interaction between this matricellular protein and αvβ3 integrin is suggested to promote the development of peritoneal metastasis [182]. Coincidentally, this interaction allows the binding of CAV1 to this complex between Cyr61 and αvβ3 integrin in bronchial epithelial cells [183]. In addition, S100 proteins are a family of calcium-binding proteins, which are found together with caveolins as cargos in exosomes derived from metastatic hepatocellular carcinoma samples [121]. The presence of S100 correlates with a diminished overall survival in patients with breast cancer [184,185], and is held responsible for promoting the homing or targeting of tumor cells to pre-metastatic niches [186]. On the other hand, presence of tenascin, an extracellular matrix glycoprotein, is associated with stromal levels of CAV1 in patients with non-small cell lung cancer [187], and is thought to enhance the viability of breast cancer cells that initiate metastasis and has been described as an extracellular matrix constituent of the metastatic niche [188,189,190]. Additionally, Tenascin-C promotes migration and anchorage independent growth of non-metastatic breast cancer cells, such as MDA-MB-435 [191], and also regulates tumor angiogenesis by controlling the expression of vascular endothelial growth factor (VEGF) [192].

## 5. Concluding Remarks 

In summary, CAV1 is implicated in cancer, both as a promoter and a suppressor of this disease (see Figure 1). Multiple mechanisms explaining this duality are discussed here. Taken together, they point towards a “complex” role for CAV1 in cancer development, because on the one hand the protein promotes completely opposing facets of the disease, and on the other because the protein appears to modulate so many different pathways and do so often in a cell context-specific fashion. Our view is that these transitions in function are associated with the presence or absence of certain proteins depending on disease progression. Specifically, we have shown that tumor suppression by CAV1 is linked to the expression of E-cadherin. Upon E-cadherin down-regulation, as occurs during epithelial–mesenchymal transition, CAV1 takes on a completely different role by promoting signaling pathways associated with enhanced migration and invasion. A large part of this review focused then on discussing the role of CAV1 in such processes and metastasis (Table 1). A number of different mechanisms are summarized, illustrating how CAV1 promotes such traits upon expression in cancer cells (intrinsic mechanisms) (Figure 1). However, it has become apparent that CAV1 is also included into EVs, and evidence is discussed showing that CAV1 plays a significant role in determining cargo composition of EVs. Finally, the ability of CAV1 containing EVs from metastatic cells to promote malignant traits in more benign recipient cells (extrinsic mechanisms) appears, at least in part, attributable to the transfer of specific cargos present due to CAV1 rather than the transfer of CAV1 itself. Future research in this area of extrinsic mechanisms is likely to uncover many new and surprising insights to the biology of this fascinating protein.

## Figures and Tables

**Figure 1 biomolecules-09-00314-f001:**
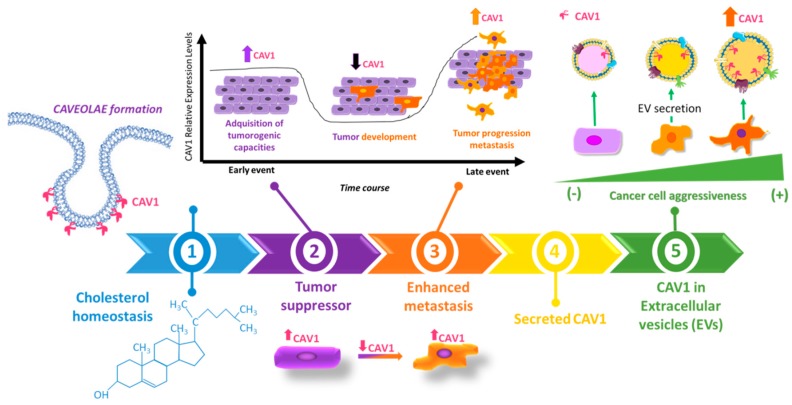
Time line summarizing the evolution of our understanding of Caveolin-1 (CAV1) function. (1) Early studies indicated at the onset of the time line centered on the structural role of the protein and its function in cholesterol transport. (2) Then, several studies emerged relating CAV1 presence to suppression of oncogenic signaling and correlating cell transformation with loss of CAV1 expression. (3) In later stages of cancer, elevated CAV1 protein levels are often detected and associated with a more malignant (metastatic) cell phenotype, indicating that in this context, CAV1 regulates different cellular traits. Mechanisms considered to this point are linked to CAV1 function within the cell, referred to here as being “intrinsic”. (4) CAV1 was then identified as a secreted protein and “extracellular” presentations of the protein are described. (5) Amongst those, one that is gaining enormous interest currently relates to its possible function(s) in extracellular vesicles (EVs), vesicular nanocarriers of cancer disease.

**Figure 2 biomolecules-09-00314-f002:**
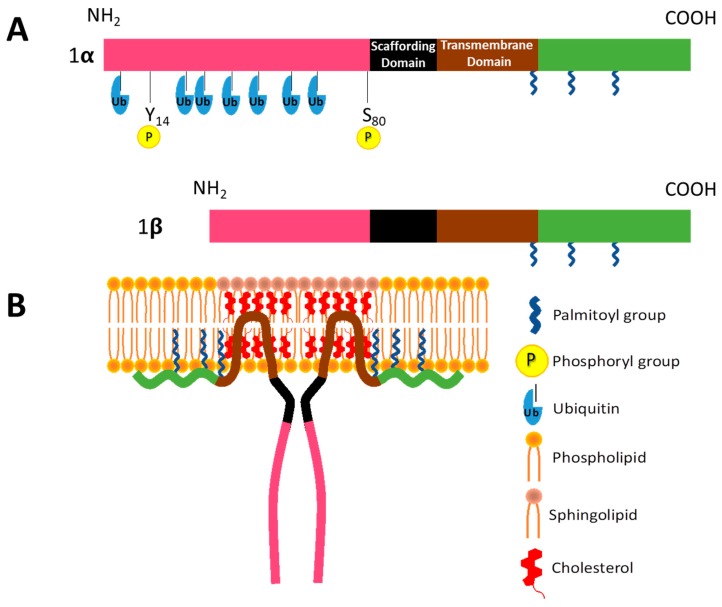
Proposed structure of Caveolin-1 (CAV1). (**A**) Schematic of CAV1α highlighting different domains and posttranslational modifications, including phosphorylation on tyrosine 14 and serine 80, ubiquitination of N-terminal residues and carboxyterminal palmitoylation sites. Simplified schematic for the β isoform. (**B**) CAV1α without amino-terminal modifications partially inserted into sphingolipid and cholesterol-enriched regions via hairpin-like membrane insertion domain.

**Table 1 biomolecules-09-00314-t001:** Summary of evidence linking CAV1 expression to cancer metastasis. Note that the data summarized here represents an update on related information provided in a table published in [29].

Model	Study Type	Major Finding	Reference
Embryonal rhabdomyosarcoma	In vivo/in vitro	Cav-1 overexpression enhances tumor formation and metastasis to the lung	[75]
Human breast carcinoma cells MDA-MB-231	In vitro	CAV1 was overexpressed in low shear stressed cells and prevented tumor cells from anoikis, while depletion of CAV1 restored sensitivity to anoikis.	[76]
Lung cancer	In vivo/in vitro	CAV1 and STAT3 are involved in electrotaxis playing a role in cell migration guidance	[77]
Human colorectal cancer	In vivo/in vitro	CAV1 ubiquitylation and subsequent degradation is promoted by NDRG1, which inhibits Epithelial-Mesenchymal transition (EMT), migration and invasion	[78]
Clear cell renal cell carcinoma	In vitro/clinical	CAV1 is overexpressed in renal cell carcinoma and has diagnostic and prognostic value. In vitro, it promotes cell migration and invasion	[79]
Hepatocellular carcinoma	In vitro/in vivo/clinical	Hypoxia upregulates CAV1 expression, which acts on calcium-binding protein S100P and promotes metastasis	[80]
Melanoma	In vitro/in vivo	CAV1 is phosphorylated on tyrosine-14 in an extracellular matrix-specific manner, and this is required to promote melanoma	[71]
Ewing sarcoma	In vitro/in vivo	CAV1 regulates MMP-9 expression through MAPK/ERK pathway, in this way regulating Ewing’s sarcoma metastasis	[81]
Breast carcinoma MDA-MB-231 cells	In vitro/in vivo	CAV1 is mechanosensitive to low shear-stress exposure, and its activation induces PI3K/Akt/mTOR signaling, which promotes motility, invadopodia formation and metastasis	[82]
Uveal melanomas	Clinical	High expression of CAV1 is associated with metastatic disease, larger tumor size, lymph node metastasis and invasion of the optic nerve head	[83]
Prostate cancer	In vitro	Non-caveolar CAV1 enhances lymphatic endothelial cell proliferation, migration and differentiation, thus promoting lymphagiogenesis	[84]
Hepatocellular carcinoma	In vitro	CAV1 inhibits autophagy, thus promoting tumor growth and metastasis	[35]
Lung adenocarcinoma	In vitro/in vivo/clinical	Overexpression of CAV-1 increased proliferation, migration and invasion. CAV1-expressing cell tumors were larger in an in vivo xenograft model. In patients, CAV1 expression correlated positively with lymph node metastasis and cancer stage	[85]
Pancreatic cancer	In vitro/in vivo/clinical	CAV1 is overexpressed in human pancreatic cancer cell lines, mouse models, and patients tumors, and is associated with worse tumor grade	[86]
Inflammatory Breast Cancer Cell	In vitro	CAV1 down regulation reduces cell invasion. Activation of Akt1 is also decreased leading to reduced phosphorylation of RhoC GTPase	[87]
Lung cancer	In vitro	CAV1 levels positively correlate with anoikis resistance, anchorage-independent growth, migration, and invasion in a variety of lung carcinoma cells	[88]
Human breast cancer BT474 cells	In vitro	CAV1 knockdown decreases cell proliferation, migration, and invasion. In addition, activity of the extracellular signal-regulated kinase 1/2 pathway was reduced. Likewise, expression of the cell cycle-associated proteins (cyclin D1, c-Fos and β-catenin), and metalloproteinases (MMP-1, -2, -9), were also decreased, while E-cadherin increased	[89]

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
