# Peer review of "Cell Intrinsic and Extrinsic Mechanisms of Caveolin-1-Enhanced Metastasis"

_biomolecules, 2019, doi:10.3390/biom9080314_

Round 1
Reviewer 1 Report
Suggestion: Change "Cancer and Metastasis" for "Introduction". It would be better to include a couple more figures, like the structure of the protein and the isoforms generated, or the role of CAV1 in vesicle transport to make the manuscript more illustrative. Also to make the manuscript more comprehensive, a table summarizing the evidence relating CAV1 and cancer and metastasis could be included. Reorganize sections, because you can find some large sections and some very short ones that could be part of a larger subject. For example, caveolins and cancer could be a separate sub-section. In the same fashion, caveolins as secreted proteins should be a complete section where the subjects concerning vesicle transport could be included. This could also be accomplished by introducing numbers to identify and separate different sections and sub-sections. The conclusion should be enriched. Minor comments: In some cases, after the references you find this: ".."Author Response
Reviewer 1:
Suggestion: Change "Cancer and Metastasis" for "Introduction".
Response: the section title was changed as suggested.
Suggestion: It would be better to include a couple more figures, like the structure of the protein and the isoforms generated, or the role of CAV1 in vesicle transport to make the manuscript more illustrative. Also to make the manuscript more comprehensive, a table summarizing the evidence relating CAV1 and cancer and metastasis could be included.
Response: We have included an additional figure (2) showing the structure of the protein. The role of CAV1 in vesicle transport was not necessarily a focus of this review, so readers were mostly referred to excellent reviews on the topic with the appropriate illustrations. Moreover, a small subsection was added to address generalities of vesicle transport (see pages 3-4). A table (1) was included relating CAV1 to metastasis with updated data since 2014, when we published a review summarizing existing data to that point (Wehinger-Nuñez et al., 2014).
Suggestion: Reorganize sections, because you can find some large sections and some very short ones that could be part of a larger subject. For example, caveolins and cancer could be a separate sub-section. In the same fashion, caveolins as secreted proteins should be a complete section where the subjects concerning vesicle transport could be included. This could also be accomplished by introducing numbers to identify and separate different sections and sub-sections.
Response: Numbers were included in each reorganized section and subsection as suggested.
Suggestion: The conclusion should be enriched.
Response: As suggested the conclusion was enriched.
Minor comments: In some cases, after the references you find this: ".."
Response: This has been corrected
Reviewer 2 Report
Minor
English editing is needed “upregulation of MMPs and which aid in the process of invasion.”
Major
Authors provide conflicting results without any interpretation and comment, but just adding one after the other. The paper requires to be re-written avoiding this list of studies with conflicting results. I give an example just to highlight this major weakness. Furthermore, without evidences from clinical setting, any statement regarding CAV1 role should be made with great caution.
“Alternatively, in clinical studies of breast cancer, nodal metastasis was found to correlate with methylation (i.e silencing) of CAV1 and CXCR4 in tumors and lymph nodes [73].”
“However, metastatic tumor cells are able to avoid anoikis upon detachment, which allows them then to invade other organs. CAV1 also participates in breast cancer metastasis by suppressing this process.”
Author Response
Reviewer 2:
Major comments: Authors provide conflicting results without any interpretation and comment, but just adding one after the other. The paper requires to be re-written avoiding this list of studies with conflicting results. I give an example just to highlight this major weakness. Furthermore, without evidences from clinical setting, any statement regarding CAV1 role should be made with great caution.
Suggestion: “Alternatively, in clinical studies of breast cancer, nodal metastasis was found to correlate with methylation (i.e silencing) of CAV1 and CXCR4 in tumors and lymph nodes [73].”
“However, metastatic tumor cells are able to avoid anoikis upon detachment, which allows them then to invade other organs. CAV1 also participates in breast cancer metastasis by suppressing this process.”
Response: We have rephrased and reorganized some sections, and added some new information that should address this concern. Confusion may have arisen because the wording used at the time was somewhat unfortunate when referring to “conflicting results”. What we wanted to indicate is that many different mechanisms have been described to explain how CAV1 promotes metastasis. In the current version this should have become clear (see pages 8, 9 and 10). Also a new Figure 2 and Table 1 was added.
Minor comment: English editing is needed “upregulation of MMPs and which aid in the process of invasion.”
Response: The text has been carefully revised by a native speaker.
Reviewer 3 Report
This an extensive review about Caveolin-1 role in metastasis. these are few points authors can address or comment.
Authors mentioned CAV1 role is controversial in metastasis, can they shed light on why these discrepancies arise
Can authors comment on CAV1 could be targeted in clinic if it's playing a role in metastasis, and summarize if there is preclinical data on targeting CAV1
Author Response
Reviewer 3:
This is an extensive review about Caveolin-1 role in metastasis. these are few points authors can address or comment.
Suggestion: Authors mentioned CAV1 role is controversial in metastasis, can they shed light on why these discrepancies arise
Response: The wording used at the time was somewhat unfortunate. What we wanted to indicate is that several different mechanisms have been described to explain how CAV1 promotes metastasis. In the current version this should have become clear (see pages 8, 9 and 10).
Suggestion: Can authors comment on CAV1 could be targeted in clinic if it's playing a role in metastasis, and summarize if there is preclinical data on targeting CAV1.
Response: We have now included a paragraph indicating how CAV1 might serve to improve diagnosis and treatment in patients (see page 10).
Round 2
Reviewer 2 Report
I do not have other comments